# The Effectiveness of Allergen Immunotherapy in Adult Patients with Atopic Dermatitis Allergic to House Dust Mites

**DOI:** 10.3390/medicina59010015

**Published:** 2022-12-21

**Authors:** Agnieszka Bogacz-Piaseczyńska, Andrzej Bożek

**Affiliations:** Clinical Department of Internal Diseases, Dermatology and Allergology in Zabrze, Medical University of Silesia in Katowice, Sklodowskiej 10, 41-800 Zabrze, Poland

**Keywords:** atopic dermatitis, allergen immunotherapy, IgE, house dust mites

## Abstract

*Background and objectives*: Allergen immunotherapy (AIT) is not a first-line therapy in atopic dermatitis (AD) and its effectiveness has been criticised. Objectives: The efficacy and safety of AIT in adult patients with AD and monosensitisation to house dust mites (HDMs) were investigated. *Materials and Methods*: A total of 37 patients were included in this double-blind, placebo-controlled study. Patients were eligible if they were diagnosed with AD; had moderate-to-severe AD according to the Eczema Area and Severity Index (EASI) with at least 7.1 points, the % BSA (body surface area) scale with at least 16 points, and the IsGA (investigator global assessment) scale with 3 points; had positive skin prick tests (SPTs); and were positive for the specific immunoglobulin E (sIgE) response to *D. pteronyssinus* and *D. farinae* extracts, as well as Der p 1 and Der f1. The patients received Purethal mites (20,000 AUeq/mL, HAL Allergy, Leiden, The Netherlands) with the extract allergens *D. pteronyssinus* and *D. farinae* (50/50%) or a placebo for 12 months. The primary outcomes included changes in EASI, % BSA, and IsGA due to SCIT between the start and after 12 months of therapy. *Results:* In the study group, significant improvement was observed in terms of the EASI score from 43 ± 8.2 to 21 ± 5.9 points, % BSA from 72 ± 18 to 28 ± 11 points, and IsGA from 4.5 ± 0.5 to 1.5 ± 0.5 points in comparison with the placebo after 1 year of AIT. Additionally, the proportion of patients who achieved success in the IsGA (IsGA < 2) was significantly better in comparison to the placebo with 13/20 (65%) vs. 4/14 (29%), respectively (*p* < 0.05). *Conclusions*: HDM-AIT effectively improved atopic dermatitis in patients that strictly qualified for desensitisation with a confirmed monovalent mite allergy.

## 1. Introduction

According to the WHO, allergies are currently ranked third on the list of the most common chronic diseases and constitute one of the severe threats to civilisation [1,2]. This is a heterogeneous group of diseases, including bronchial asthma, allergic rhinitis, food allergy, and atopic dermatitis. IgE-dependent atopic diseases include the most critical group, resulting in an inadequate immune response to familiar and potentially harmless allergens. House dust mites are one of the major allergens that play a role in most of these related diseases [1,2].

Atopic dermatitis (AD) is a genetically determined, chronic, IgE-mediated skin disorder. The prevalence of AD has increased in industrialised nations, with approximately 15% to 20% of children and 1% to 3% of adults affected worldwide. The onset of the disease commonly presents by five years of age, with the highest incidence occurring between the ages of 3 and 6 months. Approximately 60% of patients develop the disease in the first year of life and 90% within the first five years of life. A total of 20% of children who develop AD before two years of age will have persistent disease symptoms; 17% will have intermittent symptoms by seven years. Only 16.8% of adults with AD experience onset after adolescence. AD frequently resolves by the time a child reaches adulthood. However, approximately 10% to 30% of patients will continue to have symptoms of the disease [3,4]. The pathogenesis of AD is influenced by skin barrier disruption, an inappropriate immune response, and abnormal microbial skin colonisation [3,4]. Briefly, two immunological mechanisms participate in the pathogenesis of AD: type I, the IgE-dependent allergic mechanism, and type IV, the allergic mechanism dependent on the pathway of allergen presentation to T lymphocytes facilitated by specific IgE. IgE antibodies in AD patients react mainly with food and airborne allergens, for example, house dust mites. *D. pteronyssinus* and *D. farinae* can play an essential role in the pathogenesis of atopic dermatitis [3].

It needs to be emphasised that AD is also associated with cutaneous dysbiosis. Microbial inter-kingdom and host–microbiome interactions may play a critical role in the modulation of atopic dermatitis. Increased richness and abundance of Staphylococcus, Lactococcus, and Alternaria were found in atopic patients [5]. Their biofilm supernatant could influence keratinocyte biology, suggesting an additive effect on the AD-associated host response. The problem of dysbiosis requires further research.

Atopic dermatitis is a very complex disease and therefore requires different treatment approaches. Many patients experience problems throughout their lives, creating a need for long-term treatment of the disease. Emollients should be used in each patient, regardless of the severity of the disease. In more severe conditions, glucocorticosteroids, immunosuppressants, phototherapy PUVA, and allergen-specific immunotherapy are used.

Allergen-specific immunotherapy is the only causal treatment for atopic dermatitis patients; the effects persist for many years after the cessation of therapy. This therapy reduces symptoms, decreases medication use, and improves the quality of life. It can inhibit the progression of IgE-mediated diseases [6,7,8].

Opinions regarding the effectiveness of AIT in the treatment of AD are divided. The current international consensus regarding allergen immunotherapy does not recommend such treatment. However, it is not a contraindication to AIT in patients with multiform atopic disease [9].

In a meta-analysis, Tam et al. concluded that there is no consistent evidence that AIT is effective in treating AD. Still, due to the low quality of the evidence, further research is needed to determine whether AIT plays a role in the treatment of AD [10].

HDM extract contains numerous molecules characterised by high allergenicity. The molecules Der p 1, Der p 2, and Der p 23 are of the highest clinical importance. There is some evidence that house dust mite AIT is more effective in patients that are sensitive to these major extract molecules [11,12]. However, AIT against HDM needs confirmation in patients with AD.

In the present study, we wanted to check the efficacy and safety of AIT in monosensitised patients against house dust mites while checking the molecular profile of this sensitisation. We tested the hypothesis that checking the first antigen profiles of mite antigens affects the effectiveness of a particular immunotherapy.

## 2. Material and Methods

### 2.1. Study Design

This randomised, placebo-controlled, double-blind 12-month trial was performed in the Clinical Outpatient Allergy Department between 2021 and 2022. All included patients had received 12 months of allergen injection immunotherapy (SCIT) with the use of an HDM allergen extract or placebo.

### 2.2. Patients

In the initial phase of the study, 129 adult patients were screened for clinical symptoms suggestive of AD. A dermatologist and an allergist evaluated these patients. Seventy-one patients with a final AD diagnosis were subjected to further detailed analysis. Finally, 37 patients who met all the criteria were included in the study. Patients were eligible if they were diagnosed with AD a minimum of one year before the study (documented one year of therapy of AD); were between 16 and 35 years of age; and had moderate-to-severe AD symptoms according to the following assessments: at least 7.1 points on the Eczema Area and Severity Index (EASI), at least 16 points on the % BSA (body surface area) scale, and at least 3 points on the IsGA (investigator global assessment) scale; had a positive skin prick tests (SPT); were positive for specific immunoglobulin E (sIgE) in response to extracts of *D. pteronyssinus* and *D. farinae*, as well as to Der p 1 and Der f1; and had negative results of SPT and sIgE to other inhalant allergens.

The exclusion criteria were as follows:Other active skin diseases;Systemic immunosuppressant treatment (oral corticosteroids, cyclosporine, methotrexate) during the last 12 months;Other chronic diseases;Lack of written consent provided.

The characteristics of the patients are presented in Table 1 and the flow chart in Figure 1.

### 2.3. Scales

The EASI assesses disease extent on a scale of 0 to 6 in 4 defined body regions plus an assessment of erythema, infiltration, and/or population; excoriation; and lichenification, each on a scale of 0 to 3. A formula is then used to calculate the total score for each of the 4 regions, which are then added together. Interpretation of the EASI result: 0 = no change, 0.1–1.0 = almost no change, 1.1–7.0 = mild in intensity, 7.1–21.0 = moderate intensity, 21.1–50.0 = high intensity, and 50.1–72.0 = very severe.

The IsGA is a doctor-assessed outcome that evaluates overall AD severity on a 5- or 6-point scale ranging from clear (0) to severe (4) or very severe (5).

In this analysis, % BSA was categorised according to severity bands: clear (0%), mild (>0 to <16%), moderate (16 to <40%), and severe (40–100%).

The dermatologist assessed signs of AD at each study visit.

### 2.4. Treatment

In the study group, patients received Purethal mites (20,000 AUeq/mL, HAL Allergy BV, Leiden, The Netherlands) with the extract allergens *D. pteronyssinus* and *D. farinae* (50/50%) or a placebo. Purethal was administered as a perennial therapy using the following regimen: 1 dose—0.1 mL, 2 doses—0.2 mL, 3 doses—0.5 mL every week, and 0.5 mL every four weeks. The average cumulative dose was 305,000 BAU (bioequivalent allergy units) administered to each patient undergoing active treatment. Patients in the placebo group received placebo injections without extract allergens that were prepared by the same manufacturer (HAL Allergy BV, Leiden, The Netherlands). Each patient, regardless of their group, received 16–17 injections during the year.

In the study and placebo group, oral antihistamines, such as desloratadine and topical medications, were added depending on individual diseases. During observation, patients with clinical signs of bacterial skin infection, such as erosions with honey-coloured crusts, were allowed to receive therapy with topical Bactroban (mupirocin ointment) and/or a 7-day course of amoxicillin and/or prednisolone 0.5 mg kg over 10 days that was administered on any occurrences of skin exacerbation, including superinfection. All patients always used emollients.

Symptomatic treatment was scored as a medication score: one point each for desloratadine, mometasone furoate cream, and mupirocin ointment use each day; 7 points for each course of amoxicillin; and 10 points for a course of encortolon. The patients were required to record symptomatic drug use in the diary card.

### 2.5. Randomisation Procedure

Thirty-seven patients were individually randomised in comparable numbers to one of two “parallel” groups using a double-blind method (Figure 1). The randomisation procedure with random selection relied on the use of computer-generated numbers by the use of a coin flip generator (Excel, 2021, Microsoft). Finally, 21 subjects received SCIT, and 16 participants formed the placebo group. The groups were comparable at baseline in most of the variables (Table 1).

### 2.6. Diagnostic Procedures

Careful examination of the eyes, ears, nose, and dermatological features was performed on all patients.

### 2.7. Skin Prick Tests

The SPT was performed using inhalant allergens (HAL Allergy B.V, Leiden, The Netherlands) from the following panel: *D. pteronyssinus*, *D. farinae*, 5 mixed grasses (*Phleum pratense*, *Dactylis glomerata*, *Anthoxanthum odoratum*, *Lolium perenne*, and *Poa pratensis*), mixed tree, mugwort, Alternaria, Cladosporium, and dog and cat allergens. Positive (10 mg/mL of histamine) and negative (saline) controls were also included. A house dust mite allergy was defined as a positive skin test for *D. pteronyssinus* and *D. farinae* allergens with a minimum wheal diameter 3 mm greater than the negative control. Patients with negative tests for histamine sensitivity were excluded from further analyses.

### 2.8. Clinical Assessment

In all included patients, clinical assessments with the use of EASI, % BSE, and IsGA scales and the Dermatology Life Quality of Index (DLQI) were performed at the start and after 12 months of treatment.

The Dermatology Life Quality Index (DLQI) is a ten-question questionnaire used to measure the impact of skin disease on the quality of life of an affected person. It is designed for people aged 16 years and above.

### 2.9. Laboratory Tests Included

The concentration of total IgE in blood serum, specific IgE and IgG4 in response to the extracts of *D. pteronyssinus* and *D. farinae*, and specific IgE in response to components Der p1 and Der f 1 were determined using Immuno CAP (ThermoFisher Scientific, Uppsala, Sweden) following the manufacturer’s instructions. The results were considered positive when the sIgE concentration was more significant than 0.35 IU/mL.

### 2.10. Outcomes

Primary outcomes included changes in EASI, % BSA, and IsGA due to SCIT between the start and after 12 months of therapy; the proportion of patients who achieved success in ISGA (defined as ISGA of clear [0] or almost clear [1] with ≥2-grade improvement from baseline); and a reduction in medication score. We also analysed the number of exacerbations of AD (as a need to use systemic steroids).

Secondary outcomes were the improvement in DLQL score, changes in sIgG4 and sIgE in response to the extracts of *D. pteronyssinus* and *D. farinae*, and sIgE in response to Der p1 and Der f1 after 12 months.

The statistical analysis was performed using Statistica version 8.12 (SoftPol, Cracow, Poland). The non-parametric tests were used because the data were not normally distributed. The Wilcoxon test was used to analyse differences between the groups. The ANOVA test was used to compare scale scores. Differences were considered significant when *p* < 0.05.

## 3. Results

### 3.1. Clinical Improvement and Medication Score

In the study group, significant improvement was observed on the basis of EASI, % BSA, and IsGA scales compared with the placebo (Table 2). Additionally, the proportion of patients who achieved success in IsGA was significantly better in comparison to the placebo with 13/20 (65%) vs. 4/14 (29%), respectively (*p* < 0.05). Patients that underwent SCIT reported a more significant improvement in quality of life according to the DLQI from 11.76 ± 2.31 to 7.03 ± 1.82 (*p* < 0.05) in comparison to the placebo, which went from 12.21 ± 1.22 to 10.56 ± 2.34 (*p* > 0.05). During the 12 months of observation, the number of AD exacerbations in the study group was significantly lower compared with the placebo: 1.4 vs. 2.9 per patient during 12 months of observation for *p* < 0.05. The medication score was significantly decreased only in the AIT group. The data are presented in Figure 2.

### 3.2. Immunological Markers

Serum-specific IgE against *D. pteronyssinus, D. farinae*, D pter 1, and D far 1 decreased during the SCIT therapy (Figure 3). Serum-specific IgG4 against *D. pteronyssinus* and *D. farinae* increased after 12 months of immunotherapy in the study group (Figure 4). The concentration of serum IgG4 in the placebo group was constant against the analysed allergens at a very low level.

One mild systemic reaction requiring the administration of an antihistamine (transient urticaria) was reported during the immunotherapy. In addition, a mild local reaction was noted for 13 (0.04%) injections in 5 patients in the study group. These reactions resolved spontaneously up to 60 min after the injection. There were no systemic or local reactions in the placebo group.

## 4. Discussion

Herein we present the results of a randomised, double-blind, placebo-controlled trial that assessed the efficacy of HDM SCIT therapy after 12 months in adult patients monosensitised to mites. A significant reduction in the EASI confirmed that the IsGA and % BSA scales were tested, which was consistent with other, single observations. Contrary to many studies, we decided to perform a comprehensive quality-of-life assessment using the DQLI and did not analyse individual factors influencing well-being, e.g., testimony [13,14,15]. We found a significant improvement in the quality of life in the desensitised group and, albeit significantly less, in the placebo group. This can be explained by the placebo effect mentioned earlier. In our opinion, the multi-level assessment of the quality of life and not just one parameter reduces the subjectivity of such an assessment and increases its credibility. This, however, requires studies on a larger group of patients. In the available studies, most patients analysed only their symptoms using the VAS scale and they decreased to a greater extent in desensitised patients [16,17].

Not many studies evaluated the effectiveness of HDM allergen injection immunotherapy in adult AD patients [13,14]. One study confirmed a significant clinical and SCORAD improvement in adult AD patients after treatment with HDM SCIT [13,14]. In the presented observations, which also met similar criteria for randomisation and double-blind application, a definite improvement was obtained in most patients that underwent SCIT. It is worth emphasising that the observed improvement was more significant than in the study with sublingual HDM immunotherapy in children [15].

Another critical topic is the evaluation of the effectiveness of atopic dermatitis treatment using various scales. In most studies, the SCORAD scales without other scales were used to monitor the skin disease, which makes simple comparisons difficult. However, EASI has been preferred because disease severity grading is based on its average degree in each region rather than the selection of the representative lesion. The EASI also allocates greater weight to the disease extent. It is a primary score index that evaluates the efficacy of biological treatment for AD. The advantage of the presented work is a comprehensive skin assessment on several scales simultaneously. The study showed a significant improvement in all scales in the vast majority of patients after 12 months of SCIT compared with a placebo, where such an improvement was obtained in only one person. The greater effectiveness of HDM SCIT compared with HDM SLIT in the studied patients may result from worse compliance, which is typical in sub-oral immunotherapy, as well as the lower cumulative dose of the allergen [13,14,15].

An important essential concern is the safety of the treatment. The often-emphasised lower security of SCIT relative to SLIT was not confirmed in this work [16]. There was only one mild systemic reaction in the entire study, where mild urticaria appeared within 10 min of injection and resolved spontaneously within an hour. The remaining local responses did not require treatment, and their frequency was comparable to other clinical trials. It seems that the experience of medical personnel and the quality of the preparation used are of crucial importance in this topic.

The available studies highlight the best effect of immunotherapy on HDM in patients with mild-to-moderate AD and a weak effect in patients with severe AD [13,14,17]. However, in our study, most patients with a severe form of the disease also achieved significant clinical improvement after 12 months of HDM SCIT. This can be explained by the fact that there were rigorous inclusion criteria that involved patients that were monosensitised to mites. It can therefore be assumed that it was HDM-driven AD. This, of course, requires further research. It may also be of crucial importance that all patients were allergic to Der p1 and Der f1, which makes them likely to respond to immunotherapy. In a large group of AD patients allergic to HDM extract, immunotherapy is ineffective when there is only an allergy to Der p10. Therefore, it seems crucial to check it against the criteria for the inclusion of additional treatments.

Evidence of the effectiveness of HDM SCIT in patients with various degrees of AD severity is the appearance of IgG4 antibodies to desensitised allergens *D. pteronyssinus* and *D. farinae*. Their significant increase after one year of treatment and the lack of it in the placebo group demonstrated the effectiveness of the therapy, as in patients with other forms of the disease such as allergic rhinitis and/or asthma.

The presented observation also included patients with the aforementioned accompanying diseases, but they were not the essential criterion for inclusion in the treatment. The group of patients with atopic dermatitis and accompanying respiratory symptoms did not differ significantly in terms of immunological parameters and effectiveness after immunotherapy compared with patients with atopic dermatitis alone. However, all the first patients improved in terms of reducing their runny nose and/or asthma symptoms. Additionally, the decrease in sIgE in response to all tested allergens in the immunotherapy group confirmed the effectiveness of the treatment, although it was of secondary importance. Due to the small number of patients, these data were not presented.

The limitation of the study was the relatively small group. It resulted from restricted criteria, namely, only monosensitised HDM patients were included. The presence of polysensitisation or sensitisation to an extract of desensitised mites in the absence of the main antigenic components Der p 1 and Der f 1 may result in the ineffectiveness of such treatment, as was confirmed in other studies [18,19]. This shows that AIT in AD can only make sense in a very narrow group of AD patients. This requires further research. A limitation of the work was also the short duration of treatment; however, as in most studies with AIT, such an assessment is performed already after the first year of treatment [13,14]. Despite the homogeneity of the sensitisation of the studied patients, it is possible that not all patients had such a homogeneous disease mechanism. Therefore, not all of them achieved the total treatment effect. This has not been investigated in more detail. It should be also emphasised that the placebo effect in this observation may be significant; other authors also confirmed this [13,17]. In the presented study, an improvement was also noted according to the placebo rupee in two cases, which was significant despite using only symptomatic treatment. However, the use of a double-blind regimen minimised the placebo effect in this trial and hence there was a good effect in the study group.

An alternative to ongoing treatment may be dupilumab. It is a human monoclonal antibody that is directed against the alpha subunit of the interleukin-4 receptor and inhibits the signalling of IL-4 and IL-13 as an IL-4/IL-13 receptor blocker. Dupilumab is approved for treating asthma and other type-2 inflammatory diseases [20]. Many studies have confirmed the effectiveness of dupilumab, but they have not yet been associated with AIT [20,21]. Dupilumab improved clinical symptoms and quality of life in adults and children/adolescents with these diseases. However, there is a conflict in opinions regarding the safety of dupilumab. Interestingly, IL-4/IL-13 blockers unexpectedly protect against humoral autoimmune diseases but dynamically skew immune responses towards certain diseases related to the IL-23/IL-17 cytokine pathway. The IL-4/13 axis also plays a role in homeostatic tissue repair, and we have noted evidence of association with ocular and arterial pathology [22].

## 5. Conclusions

Allergen immunotherapy in patients with AD and those with monosensitised HDM significantly improved the clinical course of the disease after one year of treatment compared with the placebo group. More research is needed.

## Figures and Tables

**Figure 1 medicina-59-00015-f001:**
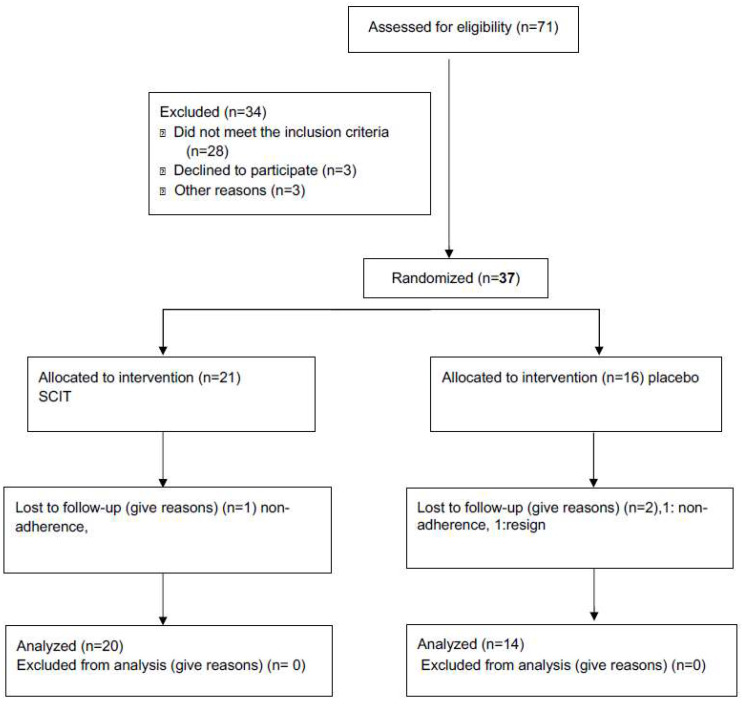
Number of participants assessed for eligibility that completed the study.

**Figure 2 medicina-59-00015-f002:**
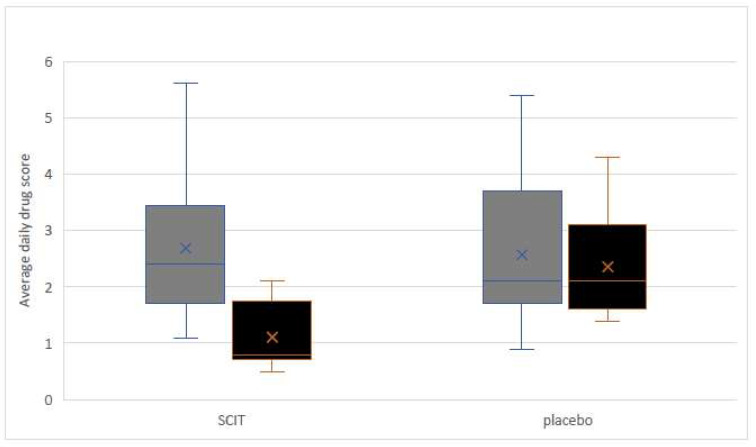
Changes in the medication score during the 12-month observation. SCIT—the group that underwent immunotherapy; * significant reduction of average daily drug score after 12 months of immunotherapy (*p* < 0.05); no significant changes in the placebo group.

**Figure 3 medicina-59-00015-f003:**
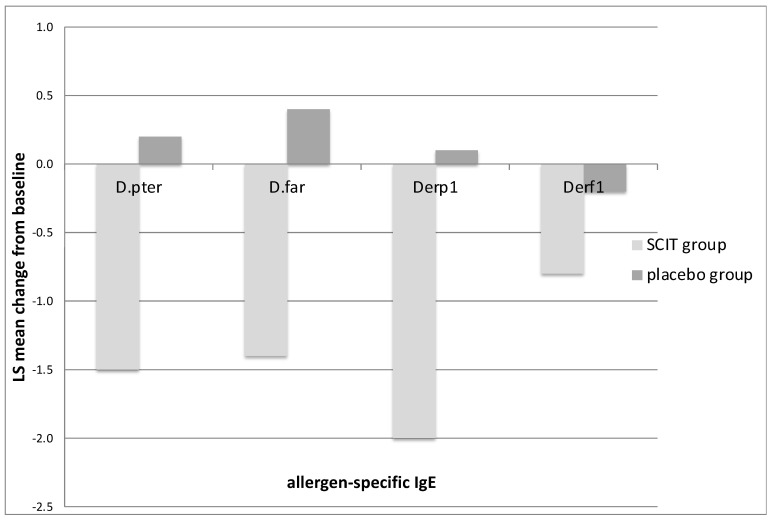
Decrease in HDM-specific IgE levels (SE) compared with the placebo at 12 months after the start of treatment. *D. pter*—mean change in IgE against *D. pteronyssinus* from baseline after 12 months of injection immunotherapy (SCIT) or a placebo; *D. far*—mean change in IgE against *D. farinae* from baseline after 12 months of SCIT or a placebo; Derp1—mean change in IgE against antigen D pter 1 from baseline after 12 months of SCIT or a placebo; Der f1—mean change in IgE against antigen Der f1 from baseline after 12 months of SCIT or a placebo. There were significant changes between the active and placebo groups in all analysed parameters.

**Figure 4 medicina-59-00015-f004:**
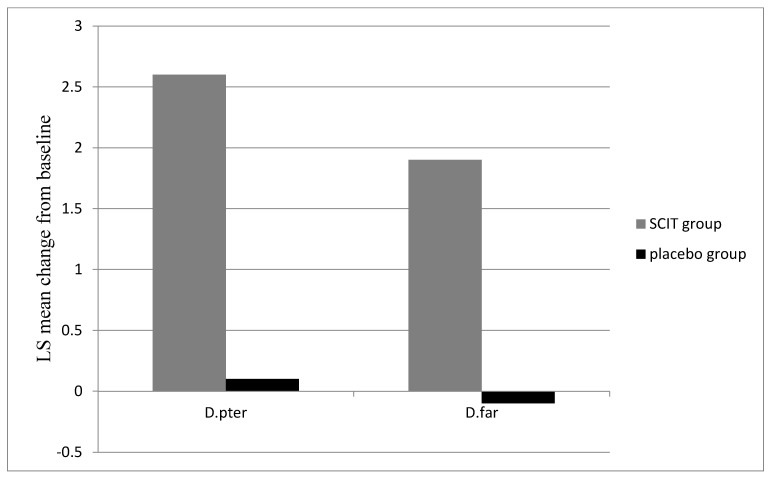
Increase in HDM-specific IgG4 levels (SE) after 12 months from the start of treatment in the SCIT group. *D. pter*—mean change in IgG_4_ against *D. pteronyssinus* from baseline; *D. far*—mean change of IgG_4_ against *D. farinae* from baseline. Treatment safety assessment.

**Table 1 medicina-59-00015-t001:** Characteristic of study patients.

	SCIT Group *n* = 21	Placebo Group *n* = 16	*p*
Women	10	8	NS
Mean age (years) SD	18.5 ± 11.3	20.1 ± 8.3	NS
Mean EASI SD	43 ± 8.2	39 ± 10.3	NS
Range	20–58	12–62	
Mean % BSA SD	72 ± 18	79 ± 21	NS
Range	23–84	19–88	
Mean IsGA SD	4.5 ± 0.5	4 ± 0.75	NS
Range	3–5	3–5	
Duration of AD (yrs)	18.6 ± 5.7	21.9 ± 9.4	0.05
Number of subjects with asthma	4	3	NS
Number of patients with allergic rhinitis	8	7	NS
Number of smokers	4	2	NS
Total IgE (kU/L) SD	7287.43 ± 1664	6450 ± 2018	0.04
Total IgE (kU/L) SD in patients with AD and IA	6933.12 ± 2070	6201.1 ± 2652	NS
Total IgE (kU/L) SD in patients with only AD	7356.4 ± 1980	6953.5 ± 2542	NS
sIgE to *D. pter* (kU/L) SD	25.1 ± 6.34	21.5 ± 9.11	NS
sIgE to *D. pter* (kU/L) SD in patients with AD and IA	23.4 ± 9.11	19.4 ± 7.21	NS
sIgE to *D. pter* (kU/L) SD in patients with only AD	27.4 ± 6.2	23.8 ± 5.19	NS
sIgE to *D. far* (kU/L) SD	13.04 ± 4.71	11.9 ± 5.2	NS
sIgE to *D. far* (kU/L) SD in patients with AD and IA	16.4 ± 5.3	9.31 ± 4.2	0.03
sIgE to *D. far* (kU/L) SD in patients with only AD	11.2 ± 4.5	15.4 ± 8.3 **	NS
Treatment before study *			
Antihistamines	21 (100%)	16 (100%)	NS
Topical glucocorticosteroids	17 (80%)	14 (88%)	NS
Calcineurin inhibitors topically	11 (52%)	9 (56%)	
			NS
Systemic glucocorticosteroid	4 (19%)	3 (18%)	
			NS
Cyclosporine	13 (62%)	9 (56%)	NS
Methotrexate	1 (5%)	0	NS
Dupilimab	0	0	NS
PUVA	7 (33%)	3 (19%)	0.03

* Therapy in the last 12 months; ** significant difference between patients with AD and IA in comparison to patients with only AD; AD—atopic dermatitis, IA—concomitant respiratory symptoms of allergy, SD—standard deviation, sIgE—allergen-specific IgE, NS—not significant, Der p—*D. pteronyssinus*, Der f—*D. farinae*, EASI—Eczema Area and Severity Index, BSA—body surface area, IsGA—investigator global assessment.

**Table 2 medicina-59-00015-t002:** The mean changes in EASI, % BSA, and IsGA in the studied patients during the 12 months of observation.

	SCIT Group *n* = 20			Placebo Group *n* = 14		
	Baseline	After 12 m	*p*	Baseline	After 12 m	*p*
Mean EASI ± SD Range	43 ± 8.2 20–58	21 ± 5.9 0–54	<0.05	39 ± 10.3 12–62	32 ± 12.8 10–69	NS
Mean EASI ± SD in patients with AD and IA Range	45 ± 6.8 22–61	24 ± 4.1 0–49	<0.05	38 ± 9.4 11–58	30 ± 9.12 11–71	NS
Mean EASI ± SD in patients with only AD Range	39 ± 7.2 18–55	18 ± 5 0–61	<0.05	41.2 ± 3 14–65	35 ± 11.8 8–65	NS
Mean % BSA ± SD Range	72 ± 18 23–84	28 ± 11 0–78	<0.05	79 ± 21 19–88	69 ± 21 19–88	0.05
Mean % BSA ± SD in patients with AD and IA Range	70 ± 9.4 20–81	26 ± 9.5 0–67	<0.05	77 ± 6.8 15–91	67 ± 14.8 14–90	NS
Mean % BSA ± SD in patients with only AD Range	75 ± 11.2 25–83	30 ± 11.5 0–78	<0.05	80 ± 12.3 10–87	70.1 ± 4.1 17–91	0.05
Mean IsGA ± SD Range	4.5 ± 0.5 3–5	1.5 ± 0.5 0–5	<0.05	4 ± 0.75 3–5	3.5 ± 0.5 1–5	NS
Mean IsGA ± SD in patients with AD and IA Range	4.5 ± 0.25 3–5	1.75 ± 0.5 0–5	<0.05	4 ± 0.25 3–5	3.5 ± 0.75 1–5	NS
Mean IsGA ± SD in patients with only AD Range	4.5 ± 0.75 3–5	1.5 ± 2.5 0–5	<0.05	3.75 ± 0.75 3–4.75	3.5 ± 0.25 1–5	NS

AD—atopic dermatitis; IA—concomitant respiratory symptoms of allergy; NS—not significant in the ANOVA test; SCIT—allergen immunotherapy; EASI—the Eczema Area and Severity Index; BSA—body surface area; IsGA—investigator global assessment; SD—standard deviation.

## Data Availability

The data presented in this study are available on request from the corresponding author. The data are not publicly available due to ethical restrictions.

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
