# Peer review of "The Effectiveness of Allergen Immunotherapy in Adult Patients with Atopic Dermatitis Allergic to House Dust Mites"

_medicina, 2022, doi:10.3390/medicina59010015_

Round 1

Reviewer 1 Report

In this paper, the authors report the results of a study in patients with atopic dermatitis and a sensitization to house dust mites. In the SCIT-treated group 12/21 (57%) also had respiratory disease, while in the placebo-treated group it was 10/16 (62%). Therefore, only in 9 SCIT-treated and 6 placebo-treated patients the presence of specific IgE toward house dust mites could be causative of Atopic Dermatitis (AD), although it would have to be proven.

I would like to know.

1)    The concentration of total IgE and specific IgE of patients with and without respiratory disease in both SCIT and placebo groups

2)    The BSA before and after treatment of patients with and without respiratory pathology in both the SCIT and placebo groups

3)    The IsGA before and after treatment of patients with and without respiratory pathology in both the SCIT group and placebo group

The authors should explain the different treatment between patients in the SCIT group and those in the placebo group (page 5 lines 111-117). They also need to be more clear about the Symptomatic treatment (page 5 lines 118-121)

Please add the number of SCIT/Placebo administrations given to each patient.

Were the drugs shown in Table 1. (Characteristic of study patients) continued during SCIT/Placebo therapy?

The authors should indicate how they procured the SCIT and how and by whom the placebo was prepared. In other words, who paid for the study?

The text is missing the results  to the safety of SCIT/Placebo.

Author Response

Reviewer 1

Thank you very much for all your comments.

We tried to incorporate all changes according to suggestions.

1)    The concentration of total IgE and specific IgE of patients with and without respiratory disease in both SCIT and placebo groups

Answer:

We added these data separately in table no 1. There were no significant differences between patients with or without respiratory. In both subgroups, atopic dermatitis was dominated disease. However, there was only a significantly higher concentration of sIgE to D.far between patients with only atopic dermatitis compared to patients with atopic dermatitis and respiratory diseases.

2)    The BSA before and after treatment of patients with and without respiratory pathology in both the SCIT and placebo groups

Answer: There were no significant differences between those subgroups of patients before and after observation. These data were added in table no2.

3)    The IsGA before and after treatment of patients with and without respiratory pathology in both the SCIT group and placebo group.

Answer: There were also no significant differences between those subgroups of patients before and after observation. These data were added in table no2.

The authors should explain the different treatment between patients in the SCIT group and those in the placebo group (page 5 lines 111-117). They also need to be more clear about the Symptomatic treatment (page 5 lines 118-121).

Answer: We agree that data about symptomatic treatment was presented unclear. All patients in the study and placebo group obtained the same symptomatic therapy according to clinical symptoms of atopic dermatitis. We also rewrite this part of the text.

Please add the number of SCIT/Placebo administrations given to each patient.

Answer: There were 16 or 17 injections of AIT per patient. This information was added.

Were the drugs shown in Table 1. (Characteristic of study patients) continued during SCIT/Placebo therapy?

Answer: The drugs shown in table 1 were used before including in the study. This information was added to the table. Then all patients used the same symptomatic therapy and AIT or Placebo

The authors should indicate how they procured the SCIT and how and by whom the placebo was prepared. In other words, who paid for the study?

Answer: Placebo was prepared by the manufacturer of the vaccine: HAL Allergy, Leiden, Netherlands. However, this study was funded by the university.

The text is missing the results for the safety of SCIT/Placebo.

Answer: There was  1 mild systemic reaction after ASIT and not many local reactions only in the study group. All data was added to the results module.

Reviewer 2 Report

The discussion and results should be more organized.  

Major corrections to the language are essential.

References should be matched with the instructions of this journal.

Author Response

Thank you very much for all your comments.

We tried to incorporate all changes according to suggestions.

The discussion and results should be more organized.  

Answer: The discussion and results module were re-ordered, and some additional information was added

Major corrections to the language are essential.

Answer: This manuscript was checked and edited by professional experts.

References should be matched with the instructions of this journal.

Answer: references were re-write according to instructions.

Round 2

Reviewer 2 Report

Good work

Author Response

.